# DAG-BASED GENERATIVE REGRESSION

## ABSTRACT

Standard regression models address associations between targeted dependent variables and selected independent variables. This paper generalizes this by proposing DAG-based generative regression as a generative process in which the model learns the data generation mechanism from real data. DAG is explicitly involved in the generative process by using structural equation models to capture the data generation mechanisms among the data variables. We learn DAG by reconstructing the model to replicate the real data distribution. We have conducted experiments to measure the performance of our algorithm to show that the results outperform the state-of-the-art by a significantly large margin.

## 1 INTRODUCTION

Regression analysis is a standard approach to estimating changes in data variables according to their statistical relations. Typically, a regression model targets a dependent variable (endogenous) and involves feature selection (e.g. information gain, independence test, Fisher's score correlation coefficient (Fisher, 1915)) to identify independent variables (exogenous) that are directly associated with the dependent variable. However, standard regression analysis only answers questions that are restricted to the targeted dependent variable based on its associations with the independent variables.

Recently, there has been a growing interest in learning data relations represented with Directed Acyclic Graph (DAG) (Bang-Jensen & Gutin, 2002). The goal of DAG-learning is to holistically capture interplays between the variables in an entire dataset, which extends standard regression analysis that hypothesizes a fixed many-to-one DAG structure between the independent and dependent variables (see more details in Section 3). Especially, in standard regression analysis, the presence of multicollinearity (i.e. high correlation /dependence among the independent variables) may cause inaccurate estimation of the effect of each independent variable on the dependent variable. These problems are addressed by DAG-based generative regression by holistically modeling the relations among all the data variables.

DAG-based generative regression models a generative process in which the model identifies the data generation mechanism from real data by reconstructing the model to replicate the real data distribution. We interpret DAG as a representation of causal relations among the data variables, which underpins how the data are generated through causation. DAG is explicitly involved in the generative process through the use of structural equation models to capture the causal mechanisms among the data variables. We fit the model to the data by generating new samples and minimizing their distribution difference against the real data samples. We explain (in Section 3) that this essentially shares the same goal as standard regression analysis, which is purposed to fit into the real data but is only limited to the (presumed) many-to-one DAG structure and additive noise distributions.

DAG-learning is strongly related to the research in causal understanding and discovery (Thulasiraman & Swamy, 1992). DAG was first introduced as part of the formal theories for causal graphs in epidemiology a very long time ago (Greenland et al., 2007), in which the absence of loops is a fundamental characteristic of causal graphs. By enforcing acyclicity, we avoid logical contradiction in causal cycles or feedback loops, where a variable's value depends on itself through a sequence of causal relationships. However, it is still not convincing that identifying a DAG structure to explain the data is equivalent to causal discovery (Vowels et al., 2021). We argue that we bring DAG-learning one step closer to true causality learning by combining DAGs with a generative process. Causality learning is more about discovering the underlying mechanisms to generate and modify the data via interventions. If we identify $X$ causes $Y$ with function $Y := f(X)$, this, in essence,

defines a generative process in which variable $Y$ is generated and changed according to variable $X$. Under the causal identifiability assumption (Neal, 2020), a causality model is capable of replicating the distribution of real data if and only if the model is correct. This suggests a strong link between causality learning and generative models that are designed to generate synthetic data samples similar to the training data they were trained on.

Causal identifiability generally refers to the ability to determine causal relationships between the variables in a given dataset. In the context of DAG-based generative regression, this is associated with the question of whether a DAG-based generative model offers a unique solution to replicate the real data, or in other words, whether we can use generated data samples from the model to measure the quality of the learned causal model - see Section 3 for more detailed analysis.

This paper makes the following contributions:

- We propose DAG-based generative regression to generalize standard regression analysis, which holistically models the entire structural relations among all the variables in a dataset to capture their conditional dependency /independency. To explain the generalization, we show (in Section 3) that standard regression analysis is tasked to learn from the distribution of the training data, but with an assumed DAG structure (i.e. many-to-one), and generative model (i.e. additive noise model).

- We also show (in Section 3) that DAG-based generative regression is capable of handling more general data distributions than the existing DAG-learning methods, which have been mainly focused on learning additive noise models through maximum likelihood estimations.

- We have conducted extensive experiments to measure the performance of our model under different settings. The results (in Section 4 and Section 5) suggest that DAG-based generative regression outperforms the state-of-the-art by a significantly large margin on both simulation and real-world datasets.

## 2 RELATED WORK

This work is closely related to DAG-learning, which is tasked to discover DAG structures from data. DAG-learning faces challenges arising from the combinatorial nature of the search space for DAG, which grows exponentially with the number of data variables (Chickering et al., 2004). Recently, DAG-learning with continuous optimization framework NOTEARS (Zheng et al., 2018) has drawn much attention evidenced by a series of NOTEARS-based DAG learning methods (Yu et al., 2019); (Lachapelle et al., 2020); (Zheng et al., 2020), which have fundamentally transformed DAG-learning from combinatorial search into a solvable continuous optimization problem through the use of acyclicity constraints.

However, DAG-learning methods are only focused on structure learning rather than discovering the underlying generative process of the data. Most of the existing methods involve only reconstruction (i.e. mean squared) loss, which is limited to the generative process with additive noise models. In comparison, DAG-based generative regression extends this to cover arbitrary data distributions and allows holistic regression analysis. This represents a step forward over both DAG-learning and standard regression analysis.

Furthermore, several approaches involving generative adversarial networks (GAN) have been proposed. One such model is Structural Agnostic Modelling (SAM) (Kalainathan et al., 2018). The method is designed to mitigate the computational and scalability limitations of the Causal Generative Neural Network (CGNN) model (Goudet et al., 2017). An acyclicity constraint, similar to the one used in NOTEARS, is applied to the adversarial training of the model. SAM produces significantly better results compared to other models available at the time, unfortunately, there is only a marginal improvement in scalability. To remedy this, several approaches have been developed including DAG-GAN (Gao et al., 2021), DAG-WGAN (Petkov et al., 2022), CAN (Moraffah et al., 2020) and CMGAN (Zhang et al., 2022). The former two are capable of handling tabular data, while the latter two work with images. All of the models scale very well and thus can discover accurate causal structures from high-dimensional data. However, most of them suffer from the same limitations. They use specific Structural Equation Models (SEM) (Chou & Bentler, 1995)

in their architectures to recover causality from their input data. This leads to accurate results when the observational data samples are synthesized using the same SEM. However, this is an unrealistic assumption that is not satisfied most of the time since real-world data may include a variety of SEMs that are not made available to the models. DAG-based generative regression produces a weighted adjacency matrix $A$ representing the implicitly learned causal graph $G$. It performs causal structure learning to discover causal relations together with the data generation mechanism under a general framework.

## 3 PROBLEM STATEMENT AND NOTATION

**Notation**: We use a vector $X \in \mathbb{R}^d$ to denote an observation with $d$ variables, $X_i, i = 1, ..., d$, and $\tilde{X} \in \mathbb{R}^d$ to denote synthetic data. $P(X)$ and $P(\tilde{X})$ are their distributions. Further, let $G = (V, E)$ denote DAGs with $d$ nodes defined in the DAG search space $\mathbb{D}$. We use a weighted adjacency matrix $A \in \mathbb{R}^{d \times d}$ to represent $G$, where $[A]_{ij} \neq 0$ indicates the existence of a weighted directed edge between vertex $i$ and $j$. $\mathcal{G}_A$ is the ground truth DAG that generates $P(X)$. $||P(X) - P(\tilde{X})||$ stands for the distance (difference) between the two probability distributions. We also use super-script to indicate a single data instance (sample), $X_i^j$ denotes the value of $X_i$ at the $j^{th}$ data sample.

**Problem statement**: Given $n$ i.i.d. observations $X$, we learn DAG (Bayesian network) $G \in \mathbb{D}$ to yield $\tilde{X}$ with distribution $P(\tilde{X})$ to match the observation distribution $P(X)$ and its underlying ground truth $\mathcal{G}_A \in \mathbb{D}$.

**Definition 3.1.** DAG-based Generative Regression: DAG-based generative regression trains the model to generate synthetic data $\tilde{X}$ with $P(\tilde{X})$ to be identical to $P(X)$.

The model involves a DAG that indicates causal and generative relations between the variables together with SEM. i.e., at each node $X_i$, there is a function $f_i : \mathbb{R}^d \to \mathbb{R}$ such that

$$\mathbb{E}[X_i | X_{pa(i)}] = \mathbb{E}_z(f_i(X, Z)), \tag{1}$$

where $pa(i)$ denotes the parents of node $X_i$ in $G$. This is based on the Markov assumption that given its parents in the DAG, a node $X_i$ is independent of all its non-descendants. $Z$ is a noise vector. The involvement of $Z$ allows $f_i$ to become a stochastic generative process to generate probability distribution $P(X_i)$.

DAG-based generative regression is a generalization from standard regression analysis. Model fitting in standard regression analysis trains model $f(X)$ from labeled data pairs $(X^j, y^j), j = 1, ..., n, X^j \in \mathbb{R}^d, y \in \mathbb{R}$ by minimizing the difference between the model output $y = f(X^j) + Z$ and $y^j$, where $Z$ is gaussian noise. In essence, $y = f(X) + Z$ is an additive noise generative model, which is a special case of Equation (1). In addition, standard regression analysis does not learn the structure between variables $X_i, i = 1, ..., d$. Instead, it takes $X_i$ as independent variables that are directly connected to $y$ which essentially presents a (fixed) many-to-one DAG structure. To this end, we view standard regression analysis as a special case of DAG-based generative regression.

DAG-based generative regression is a general framework that supports a range of data generative processes in addition to the additive noise model:

$$\tilde{X}(\tilde{X}_1, ..., \tilde{X}_d) = f(X; Z; W) \tag{2}$$

Under this general framework, an adequate metric is needed to measure $||P(X) - P(\tilde{X})||$. Previous works have explored a range of metrics that measure the distance between data distributions, e.g. Wasserstein (Arjovsky et al., 2017) and maximum mean discrepancy (MMD)(Tolstikhin et al., 2016), and so on and so forth. Some of them have been successfully applied to generative adversarial networks to generate high-quality synthetic data. They, in general, offer measurements for any distributions and hence can be leveraged by this general framework.

**Lemma 3.1.** Under the additive noise model assumption, reconstruction loss with mean squared error (MSE) $\mathbb{E}_{X, \tilde{X}}(||X - \tilde{X}||_2)$ measures distance $||P(X) - P(\tilde{X})||$.

*Proof.* The proof of lemma 3.1 is available in Appendix A.1. □

According to theory (Neal, 2020), DAG discovery from data depends on whether its underlying causal mechanism meets a set of key assumptions. In addition to the Markov and faithfulness assumptions, the identifiability assumption assumes an identifiable causal model, which suggests that there is only one unique DAG structure to generate $P(X)$. Under this assumption, we show that the model can be recovered from the data through optimization.

**Definition 3.2.** Identifiable Causal Graph Models: Under a given set of assumptions, we define the set of identifiable causal models $S_{G_A}$ as the set that contains model $G_A$ such that $S_{G_A} = \{G_A | P_{G_A}(\tilde{X}) \neq P_{G'_{A'}}(\tilde{X}), G_A \neq G_A, \forall G'_{A'} \in \mathbb{D}\}$, namely we cannot generate the same distribution with another different causal graph model $G'_{A'}$ in $\mathbb{D}$.

We prove that under the assumption of using identifiable models (e.g. nonlinear additive noise model), we can only achieve the global minimum (i.e. $P(\tilde{X}) = P(X)$) if a true causal structure is discovered.

**Lemma 3.2.** Consider $P_{G_A}(\tilde{X})$ from an identifiable causal graph model $G_A \in S_{G_A}$, and $\mathcal{G}_A$ is the true causal model. Then $G_A = \mathcal{G}_A$ if and only if $P_{G_A}(\tilde{X}) = P(X)$.

*Proof.* The proof of lemma 3.2 can be found in Appendix A.2. □

**Corollary 3.2.1.** Under Lemma 3.2, a DAG $G(V, E)$ can be recovered from a joint distribution $P(X)$ over all observations $X$.

## 4 MODEL ARCHITECTURE AND TRAINING

Without loss of generality, we use neural networks to approximate and learn $f = (f_1, ..., f_d)$. More specifically, each variable $X_i$ is modeled with a neural network of $L$ hidden layers $f_i(X, Z; W_i^1, ..., W_i^L), i = 1, ..., d$, where $W_i^l$ is the parameters of the $l^{th}$ layer. With $Z_i \in \mathbb{R}^m$ and $Z \in \mathbb{R}^{m \times d}$, $\tilde{X}$ is generated via additive noise model to approximate to distribution $P(X)$. Hence the generator function is given as follows:

$$\tilde{X}(\tilde{X}_1, ..., \tilde{X}_d) = f(X; W^1, ..., W^L) + Z \tag{3}$$

Practically, we follow the NOTEARS model architecture (Zheng et al., 2020), in which $W^1$ on the first layer are used to incorporate the adjacency matrix A (i.e. DAG). Given $X$, we learn $W^1, ..., W^L$ of the generator by minimizing $||P(X) - P(\tilde{X})||$. While theoretically both MSE (Bickel & Doksum, 2015) and Wasserstein metric offer sound data distribution metrics for additive noise models, practically they yield different results. A direct application of MSE as the reconstruction loss would collapse the noise vector $Z$ (since MSE minimizes the gap between $X$ and $\tilde{X}$). On the other hand, practically we found it hard to train a discriminator under the min-max optimization framework to guide DAG learning – normally the trained discriminator does not lead to satisfactory results for the structure learning. Hence, we propose to leverage both MSE and Wasserstein loss in the training. More specifically, MSE is used for the optimization of $W^1$ to effectively learn DAG, and $W^2, ..., W^L$ are trained with Wasserstein loss to generate realistic data without collapsing $Z$ – more details about the training with augmented Lagrangian can be found in Algorithm 1. The reconstruction (MSE) loss is given in Equation (4) and the Wasserstein loss is in Equation (5) as follows:

$$W^1 = argmin(\mathbb{E}_{X \sim P(X), \tilde{X} \sim P(\tilde{X})}[(X - \tilde{X})^2]) \tag{4}$$

$$W^2, ..., W^L = argmin(\mathbb{E}_{X \sim P(X), Z \sim P(Z)}[D_\theta(f(X, Z))]), \tag{5}$$

where $P(X)$ is the training data distribution, $Z \sim P(Z)$ is a process to sample $Z_i$ at each generator, and $P(\tilde{X})$ is the implicitly created probability distribution from our generator $f(X, Z)$. Note, the optimization in Equation (4) is subject to the acyclicity constraint as described in Zheng et al. (2020).

We conducted experiments (ablation study) with the combination of different loss terms (more results can be found in Section 5) and found the optimal configuration is to add an additional MMD term to Equation (4). Hence $W_i^1$ is learned with the following loss function:

$$W^1 = argmin(\mathbb{E}_{X \sim P(X), \tilde{X} \sim P(\tilde{X})}[(X - \tilde{X})^2] +$$
$$\mathbb{E}_{X \sim P(X), Z \sim P(Z)}[MMD(P(X), P(f(X, Z)))]), \quad (6)$$

where the 2nd term above computes the MMD between the real and fake data.

## 5 EXPERIMENTAL RESULTS

We perform a variety of tests on the proposed DAG-based generative regression algorithm with different datasets involving continuous, discrete and mixed data to evaluate the following aspects:

- Structure learning, which measures the accuracy of the learned DAG structure.

- Synthetic data quality, which examines the quality of the data from the learned generative process.

- Ablation study and sensitivity analysis to assess the configuration of the loss term and the hyper-parameter settings for the training. - for more information, the reader is referred to Appendices B and C.

Here we describe the settings of the experiments and present the results. For structure learning, our model is compared against the state-of-the-art DAG-learning methods, including DAG-WGAN (Petkov et al., 2022), DAG-Notears-MLP (Zheng et al., 2020), Dag-Notears (Zheng et al., 2018), DAG-GNN (Yu et al., 2019), GraN-DAG (Lachapelle et al., 2020), GAE (Ng et al., 2019) and VI-DP-DAG (Charpentier et al., 2022). It is important to note that, unlike our model, all the state-of-the-art models mentioned above do not involve noise terms in training. To make a fair comparison, we add noise to each competitor's architecture. The metric used throughout all experiments to measure the quality of the discovered causality is the Structural Hamming Distance (SHD) (de Jongh & Druzdzel, 2009).

---

**Algorithm 1** Training in DAG-based generative regression

**Require:** Sample $n$ observational data points $\{X^1, ..., X^n\}$ from the training data and noise vectors $\{Z^1, ..., Z^n\}$ from uniform or normal distributions. Generate $n$ synthetic data samples $\{\tilde{X}^1, ..., \tilde{X}^n\}$, where $\tilde{X}^n = f(X^n, Z^n)$.

**Ensure:** The acyclicity constraint value $h(A)$ is higher than its tolerance of error $h\_tol$ set to 1e-8. Each step during training has its own instance of the generator.

$\lambda \leftarrow 0$
$c \leftarrow 1$
$current\_h(A) \leftarrow \infty$
$k\_max\_iter \leftarrow 100$
$h\_tol \leftarrow 1e-8$
**for** k $< k\_max\_iter$ **do**
    **while** $c < 1e+20$ **do**
        **Step 1**: Learn DAG by minimizing MSE and MMD with $W^1$ of the generator (Equation 6)
        **Step 2**: Learn the generative process by minimizing the Wasserstein distance with $W^2, ..., W^L$ of the generator (Equation 5)
        **if** $h(A) > 0.25$ **then**
            $c \leftarrow c * 10$
        **else**
            $break$
        **end if**
    **end while**
    $current\_h(A) \leftarrow h(A)$
    $\lambda \leftarrow c * current\_h(A)$
    **if** $current\_h(A) \leq h\_tol$ **then**
        $break$
    **end if**
**end for**

---

### 5.1 CONTINUOUS DATA

We perform tests on the continuous data type using simulation data from prefixed structural equations and DAG structures. In each of the tests, we use 5000 observational data samples under different equations (i.e. linear $\rightarrow X = A^T X + Z$, non-linear-1 $\rightarrow X = A cos(X + 1) + Z$ and non-linear-2 $\rightarrow X = 2sin(A(X + 0.5)) + A(X + 0.5) + Z$). To test whether the model scales well, we perform experiments with datasets containing 10, 20, 50 and 100 columns. Moreover, to account for the randomness of the samples and ensure fair results, we perform each experiment 5 times and output the average SHD. The results are presented in Tables 1, 2 and 3.

Table 1: Non-parametric DAG structures recovered from linear data samples

| Model | SHD (5000 linear samples) | | | |
|---|---|---|---|---|
| | d=10 | d=20 | d=50 | d=100 |
| DAG-NOTEARS | $8.6 \pm 7.2$ | $13.8 \pm 9.6$ | $41.8 \pm 29.4$ | $102.8 \pm 53.2$ |
| DAG-NOTEARS-MLP | $4.6 \pm 4.3$ | $7.6 \pm 6.3$ | $29.6 \pm 18.5$ | $74 \pm 30.6$ |
| DAG-GNN | $6 \pm 6.9$ | $11.4 \pm 8.2$ | $33.6 \pm 21.2$ | $85.4 \pm 46.4$ |
| GAE | $5.5 \pm 4.9$ | $10.3 \pm 7.2$ | $31.3 \pm 13.8$ | $80.2 \pm 24.6$ |
| GraN-DAG | $3.4 \pm 5.2$ | $6.4 \pm 7.5$ | $25.2 \pm 14.6$ | $68.4 \pm 25.8$ |
| VI-DP-DAG | $2.1 \pm 4.5$ | $4.5 \pm 6.7$ | $22.4 \pm 12.7$ | $63.7 \pm 23.5$ |
| DAG-WGAN | $5.2 \pm 3.8$ | $9.2 \pm 5.7$ | $19.6 \pm 12.3$ | $58.6 \pm 22.7$ |
| This Method | $\mathbf{1.4 \pm 3.1}$ | $\mathbf{2 \pm 4.4}$ | $\mathbf{16.4 \pm 9.8}$ | $\mathbf{38.8 \pm 18.3}$ |

Table 2: Non-parametric DAG structures recovered from non-linear-1 data samples

| Model | SHD (5000 non-linear-1 samples) | | | |
|---|---|---|---|---|
| | d=10 | d=20 | d=50 | d=100 |
| DAG-NOTEARS | $11.4 \pm 4.5$ | $28.2 \pm 10.2$ | $55 \pm 23.1$ | $105.6 \pm 48.3$ |
| DAG-NOTEARS-MLP | $5.2 \pm 1.8$ | $15.4 \pm 4.6$ | $43.8 \pm 15.4$ | $86.2 \pm 29.8$ |
| DAG-GNN | $9.2 \pm 3.3$ | $23.4 \pm 8.4$ | $50.2 \pm 19.5$ | $98.6 \pm 37.6$ |
| GAE | $8.6 \pm 2.2$ | $20 \pm 5.7$ | $47.5 \pm 10.2$ | $92.3 \pm 18.9$ |
| GraN-DAG | $4 \pm 2.4$ | $11.2 \pm 6.5$ | $36.4 \pm 11.9$ | $72.8 \pm 21.7$ |
| VI-DP-DAG | $3.1 \pm 2$ | $9.8 \pm 5.1$ | $28.7 \pm 9.3$ | $68.1 \pm 16.5$ |
| DAG-WGAN | $6.4 \pm 1.4$ | $18.6 \pm 3.7$ | $22 \pm 8.6$ | $64.6 \pm 15.2$ |
| This Method | $\mathbf{2.6 \pm 1}$ | $\mathbf{5.2 \pm 2.8}$ | $\mathbf{18.8 \pm 6.2}$ | $\mathbf{50.2 \pm 13.4}$ |

Table 3: Non-parametric DAG structures recovered from non-linear-2 data samples

| Model | SHD (5000 non-linear-2 samples) | | | |
|---|---|---|---|---|
| | d=10 | d=20 | d=50 | d=100 |
| DAG-NOTEARS | $10.4 \pm 3.9$ | $22.4 \pm 8.1$ | $47.6 \pm 21.2$ | $112.8 \pm 57.8$ |
| DAG-NOTEARS-MLP | $5.4 \pm 1.5$ | $13.8 \pm 4.3$ | $30.4 \pm 15.7$ | $85.6 \pm 35.6$ |
| DAG-GNN | $8.4 \pm 3.2$ | $19.2 \pm 7.7$ | $36.2 \pm 18.6$ | $91.8 \pm 49.3$ |
| GAE | $7.3 \pm 1.8$ | $17.4 \pm 5.1$ | $33.7 \pm 13.7$ | $88.4 \pm 26.6$ |
| GraN-DAG | $4.2 \pm 2.1$ | $11.6 \pm 5.6$ | $25.2 \pm 14.5$ | $71.6 \pm 29.7$ |
| VI-DP-DAG | $2.8 \pm 1.6$ | $9.3 \pm 4.7$ | $23.8 \pm 13.3$ | $67.3 \pm 23.8$ |
| DAG-WGAN | $6.6 \pm 1.2$ | $15.2 \pm 3.4$ | $22.6 \pm 12.9$ | $64.2 \pm 21.5$ |
| This Method | $\mathbf{1.4 \pm 0.9}$ | $\mathbf{5.8 \pm 2.2}$ | $\mathbf{14.2 \pm 10.3}$ | $\mathbf{51.8 \pm 19.2}$ |

## 5.2 BENCHMARK EXPERIMENTS

In our experiments with discrete data, we have acquired the Child, Alarm, Hailfinder and Pathfinder benchmark datasets along with their ground truths from the Bayesian Network Repository `https://www.bnlearn.com/bnrepository`. These datasets are specifically organized for scalability testing and to ensure fair comparison against the state-of-the-art. We test our model against DAG-GNN (Yu et al., 2019) and DAG-WGAN (Petkov et al., 2022). Results are shown in Table 4.

Table 4: Non-parametric DAG structures recovered from benchmark data samples

| Datasets | Nodes | SHD | | |
|---|---|---|---|---|
| | | DAG-WGAN | DAG-GNN | This Method |
| Child | 20 | 20 | 30 | $\mathbf{17}$ |
| Alarm | 37 | $\mathbf{36}$ | 55 | 43 |
| Hailfinder | 56 | 73 | 71 | $\mathbf{63}$ |
| Pathfinder | 109 | 196 | 218 | $\mathbf{181}$ |

## 5.3 REAL DATA EXPERIMENTS

Although our experiments with simulation data demonstrate the model's capability of producing improved results, they alone are an insufficient indicator, as the simulation is different from the real world. To remedy this, we conduct experiments with a real dataset (Sachs et al., 2005), which is a widely recognized dataset by the research community consisting of 7466 samples across 11 columns, and its ground truth is estimated to contain 20 edges. The results are shown in Table 5.

Table 5: Non-parametric DAG structures from real data samples

| Model | Sachs Dataset |
|---|---|
| | SHD |
| DAG-WGAN | 17 |
| DAG-GNN | 25 |
| GAE | 20 |
| GraN-DAG | 17 |
| VI-DP-DAG | 16 |
| This Method | **9** |

## 5.4 SYNTHETIC DATA QUALITY

In this work, we have argued for the superiority of our approach over state-of-the-art models because we perform causality learning in tandem with synthetic data generation. Here, we scrutinize this claim more closely by comparing the features from a set of simulation data (d=10 nodes) with the features generated by our method. We consider the special case where our model achieves an SHD of 0 on this simulation data. We perform the following investigations to compare the real and synthetic data: computing the correlation matrices; visualizing the joint and marginal distributions; and machine learning regression analysis. Overall, our investigation shows that the synthetic data produced by the proposed framework faithfully captures the correlation structure in the data (Fig. 1), the joint and marginal distributions of the features (Fig. 2), and contains sufficient predictive information to support regression tasks (Fig. 3). Due to space limitations, we provide only a few examples for each investigation: the reader is referred to Appendix D for more results.

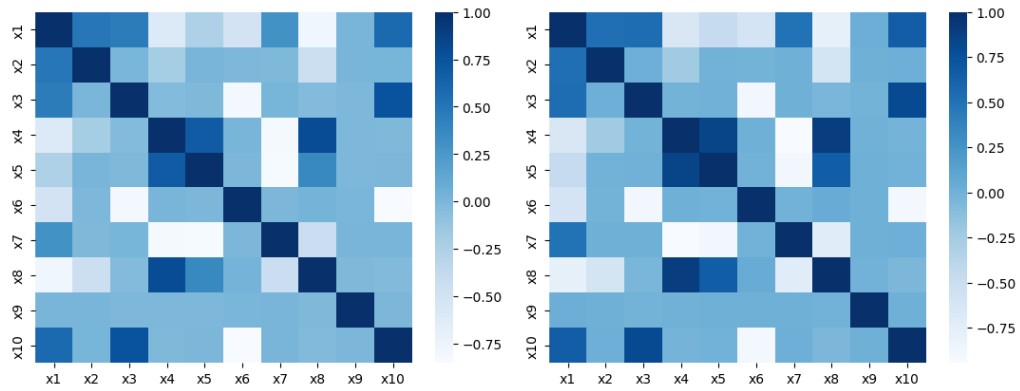

Figure 1: Comparison between the correlation matrices across the real (left) and synthetic (right) features. Our investigation shows the statistical correlations across the feature space for both the real and fake data are almost indistinguishable.

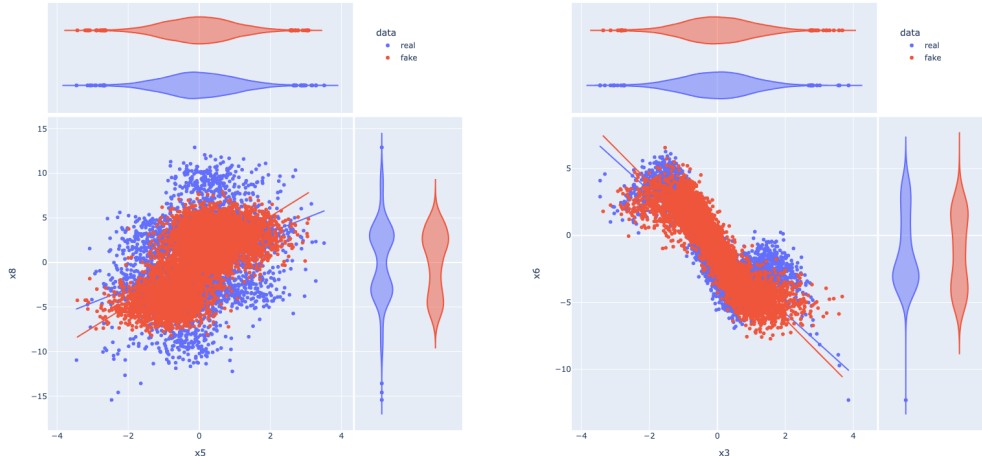

Figure 2: Visualizing the distributions of the real and synthetic features. When we plot x5 against x8 (left), and x3 against x6 (right), we observe in both cases that not only have the joint distributions been modelled correctly, but so too have the marginal distributions. Running Mann-Whitney t-tests confirms no significant differences between the real and synthetic features. More examples, including some failure cases, are available in Appendix D.

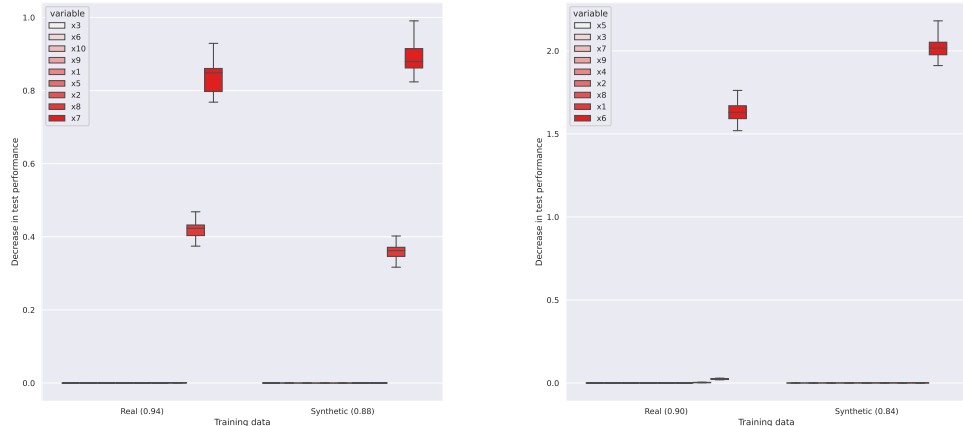

Figure 3: Using the real and synthetic data to train separate random forest (RF) regression models. When asked to predict x4 (left) and x10 (right) on the same set of hold-out test data, the synthetic RF matches the importance ordering and permutation effects of the features with the real RF, leading to similar predictive accuracy ($R^2$). Additional results are available in Appendix D.

## 6 DISCUSSION

The results presented in Tables 1, 2, and 3 reveal that the proposed DAG-based generative regression algorithm outperforms the other state-of-the-art DAG-learning methods across all data dimensionality in the three cases (linear, non-linear-1, and non-linear-2) tested in the study. Notably, the gap in SHD between our model and the others grows further in our favor with the increase in data dimensionality. This observation highlights the improvement of the performance of our proposed method in DAG-learning on datasets with a large number of variables.

The results of the benchmark experiments presented in Table 4 also suggest that our algorithm is competitive. More specifically, across all four datasets (i.e. Child, Alarm, Hilfinder, and Pathfinder), our method outperforms DAG-GNN by a significantly large margin. Furthermore, compared to the proposed method, DAG-WGAN produces inferior results over three of the four datasets. Notably, similar results are observed in the experiments with continuous datasets. Namely, the SHD gap between our method and the other methods grows more significantly with the increase of the data variable size.

So far, we have only discussed the capabilities of our model on continuous and benchmark datasets. Although the results from these experiments indicate good performance, the study would be incomplete without considering the model's ability on real-world datasets. The Sachs dataset experiment reveals that our model can discover accurate DAG structures from real data. According to the results from Table 5, our model outperforms all other state-of-the-art models by a significantly large margin.

In DAG-based generative regression, learning DAG structures from observational data goes alongside generating high-quality synthetic data. This is demonstrated by the results in Fig. 1, Fig. 2 and Fig. 3. When our model successfully recovers the ground truth from a dataset (i.e. SHD = 0), the distance between the real and fake data distributions is minimized. Moreover, all the statistical correlations in the real dataset are captured and presented in the synthetic data samples produced by our model. This shows that our model is capable of generating diverse data samples with preserved DAG structures.

The results from the ablation study and sensitivity analysis identify the combination of a set of loss terms that are optimal to our framework as well as the effect of hyper-parameter setting on the training. As we can see from Table 6, the best constellation of the loss terms in Step 1 is MSE complemented by MMD. According to Table 7, a decrease in learning and dropout rate significantly affects the performance of our model. On the other hand, increasing the size of the noise vector and the batch of the input data causes slight variations in the accuracy of the algorithm.

The outcome of the conducted experiments reveals our method can work with multiple data types (i.e. numerical, categorical and mixed) to properly recover DAG structures and produce realistic data samples. More significantly, our model outperforms the state-of-the-art DAG-learning methods. The analysis of the results indicates that the Wasserstein distance contributes to DAG learning.

All the results are produced using the additive noise model. This is a good exemplar to use as it is proven to be identifiable (Hoyer et al., 2008); (Park, 2020). However, our experiments so far are limited to this additive noise model case, which is a limitation. In future work, we will experiment with other identifiable structures, such as generalized linear models and polynomial regression.

In essence, DAG-based generative regression discovers DAG structures using a combination of MLE and adversarial loss terms with an acyclicity constraint computed through an augmented Lagrangian. As a result, our model has poor computational complexity and a sophisticated loss function. We will further experiment with more efficient structure learning frameworks and adversarial loss training for a faster model trained solely based on the Wasserstein loss.

## 7 CONCLUSION

This research has proposed DAG-based generative regression, designed as a generalization over the standard regression analysis to holistically discover DAG structures between multiple variables within a dataset and learn data generative mechanisms to generate synthetic data samples that are similar to the real data. We have conducted a theoretical analysis to explain the generalization and shown that DAG-based generative regression is capable of handling more general training data distributions. Moreover, we have demonstrated the performance of the proposed DAG-based generative regression through a series of experiments in which the proposed method outperforms the state-of-the-art DAG-learning methods by a significantly large margin. Also, our results indicate that generating high-quality data and learning DAG structures are two connected processes. Hence, producing more realistic data samples leads to the recovery of meaningful data variable relationships and vice-versa.

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

## A PROOFS

Here the reader can find the proofs associated with the Lemmas found in Section 3.

### A.1 PROOF OF LEMMA 3.1

**Lemma 3.1.** Under the additive noise model assumption, reconstruction loss with mean squared error (MSE) $\mathbb{E}_{X,\tilde{X}}(||X - \tilde{X}||_2)$ measures distance $||P(X) - P(\tilde{X})||$.

*Proof.* The additive noise model describes the generative process as follows:

$$\tilde{X}(\tilde{X}_1, ..., \tilde{X}_d) = f(X; W) + Z, \tag{7}$$

where $W$ denotes the model parameters, and $Z$ is a noise vector to the structural equation model $f(X; W)$, which explains the difference between the model prediction $\tilde{X}$ and data $X$. Hence, according to the addition term in Equation (7), one can write reconstruction error to quantify the difference between model prediction $f(X; W)$ and real data $X^j$ as follows:

$$\mathbb{E}_{X,\tilde{X}}(||X - \tilde{X}||_2) = \sum_{j=0}^{n} ||f(X; W) - X^j||_2, \tag{8}$$

where $||f(X;W) - X^j||_2$ stands for MSE between the two terms. By reducing the reconstruction loss in Equation (8) through optimization, we minimize $||P(X) - P(\tilde{X})||$. According to Peters et al. (2013), an additive noise nonlinear model is identifiable if $f$ is three times differentiable and non-linear.

Hence, in the special case of the additive noise model, MSE offers an adequate distance metric, thus concluding the proof. In fact, most of the existing works in DAG-learning assume additive noise models. □

### A.2 PROOF OF LEMMA 3.2

**Lemma 3.2.** Consider $P_{G_A}(\tilde{X})$ from an identifiable causal graph model $G_A \in S_{G_A}$, and $\mathcal{G}_A$ is the true causal model. Then $G_A = \mathcal{G}_A$ if and only if $P_{G_A}(\tilde{X}) = P(X)$.

*Proof.* As $\mathcal{G}_A \in S_{G_A}$, according to the definition of identifiable causal graph model found in Section 3, $P_{G_A}(\tilde{X}) \neq P_{G'_{A'}}(\tilde{X}), G'_{A'} \neq G_A$. Therefore, if through optimization we find $G_A = \mathcal{G}_A$, then $P_{G_A}(\tilde{X}) = P(X) \neq P_{G'_{A'}}(\tilde{X}), G'_{A'} \neq G_A, \forall G'_{A'}$, thus concluding the proof. □

## B ABLATION STUDY

We set up an ablation study to determine the best configuration of the terms in the loss function. In total, 9 experiments have been conducted on the Sachs dataset (Sachs et al., 2005) in which various combinations of the loss terms have been experimented. The first setting is labeled as "**w/o recon loss**", where no reconstruction loss is involved in training $W^1$ (namely, all $W^1, ..., W^L$ are trained with the Wasserstein loss). The rest of the results are named after the terms that we have used in the reconstruction loss for $W^1$, including reconstruction loss with MSE (Bickel & Doksum, 1977) and Negative Log Likelihood (NLL) (Grof & Transpersonal, 1921). We have also experimented with combinations of additional terms such as maximum mean discrepancy (MMD) (Tolstikhin et al., 2016) and Kullback–Leibler divergence (KLD) (Kullback & Leibler, 1951). The results of this study are presented in Table 6.

Table 6: DAG-based generative regression ablation study

| Loss function | Sachs Dataset |
| --- | --- |
| | SHD |
| w/o recon loss | 21 |
| recon loss (MSE) | 14 |
| recon loss (NLL) | 16 |
| MSE+MDD | **9** |
| NLL+MMD | 14 |
| MSE+KLD | 12 |
| NLL+KLD | 12 |
| MSE+KLD+MMD | 10 |
| NLL+KLD+MMD | 11 |

## C SENSITIVITY ANALYSIS

To ensure model robustness, we conduct sensitivity analysis to investigate how the model training responds to different hyper-parameter settings. The study is conducted to measure the accuracy of DAG reconstruction (i.e. SHD) under different hyper-parameters, which involves the following hyper-parameters: learning and dropout rate (**lr, dropout**), noise vector and batch size (**z-size, batch-size**). We start with a baseline setting as follows: **lr = 0.001, dropout = 0.5, z-size = 1, batch-size = 100**, and make changes to each of the values in question and measure the difference in SHD. All of the experiments have been conducted on the Sachs dataset and the results from them are shown in Table 7.

Table 7: DAG-based generative regression sensitivity analysis

| Hyper-parameters | Sachs Dataset |
| --- | --- |
| | SHD |
| lr = 3e-3, dropout = 0.5, z-size = 1, batch-size = 100 | 9 |
| lr = 3e-3, dropout = 0.0, z-size = 1, batch-size = 100 | 10 |
| lr = 3e-3, dropout = 0.5, z-size = 2, batch-size = 100 | 10 |
| lr = 3e-3, dropout = 0.5, z-size = 5, batch-size = 100 | 11 |
| lr = 3e-3, dropout = 0.5, z-size = 1, batch-size = 500 | 9 |
| lr = 3e-3, dropout = 0.5, z-size = 1, batch-size = 1000 | 10 |
| lr = 2e-4, dropout = 0.5, z-size = 1, batch-size = 100 | 11 |
| lr = 1e-3, dropout = 0.5, z-size = 1, batch-size = 100 | 12 |

## D ADDITIONAL RESULTS

In this section, we provide additional examples to support the analysis in Section 5.4. Specifically, we include the real-synthetic statistical comparisons for all features (Tab. 8), additional visualisations of the synthetic feature distributions (Fig. 4), and the remaining machine learning regression results (Fig. 5).

Table 8: Complete Mann-Whitney t-test results for all real and synthetic features to supplement Figure 2. We observe some failure cases, where the real and synthetic features differ significantly ($p < 0.05$).

| Feature | $p$-value |
| --- | --- |
| x1 | 7.7952e-07 |
| x2 | 0.5004 |
| x3 | 0.1683 |
| x4 | 0.0020 |
| x5 | 0.8563 |
| x6 | 0.9127 |
| x7 | 0.0364 |
| x8 | 0.1747 |
| x9 | 0.2089 |
| x10 | 6.4502e-26 |

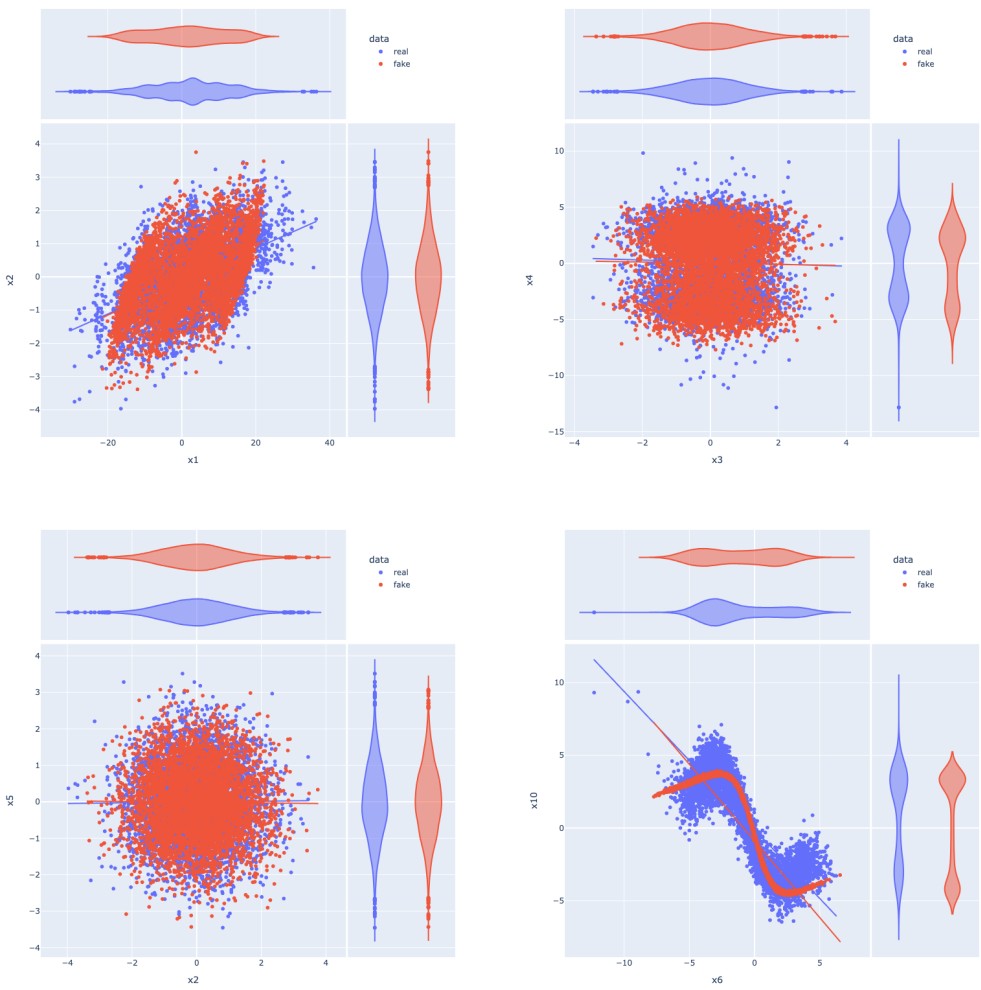

Figure 4: Further examples of the synthetic joint and marginal distributions for our method on the dataset presented in Section 5.4.

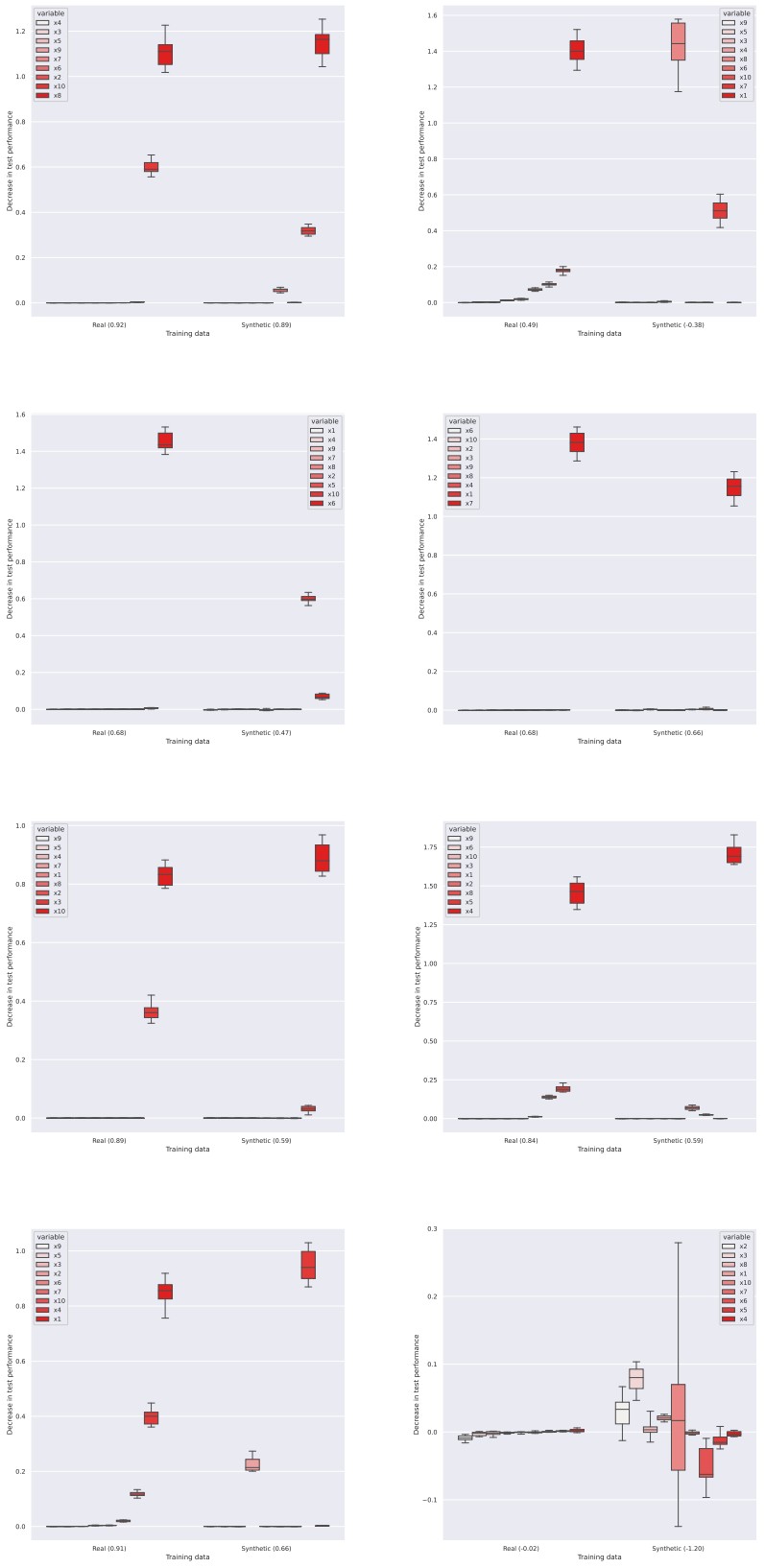

Figure 5: Remaining examples of feature importances to supplement the results in Section 5.4. We observe some failure cases, where the synthetic features differ significantly from their real counterparts.

