# OpenReview forum: "DAG-based Generative Regression"
_ICLR.cc/2024/Conference — ICLR 2024 Conference Withdrawn Submission_

### Official Review · Reviewer_ay5v · 2023-10-25

**Soundness:** 3 good
**Presentation:** 2 fair
**Contribution:** 2 fair
**Rating:** 3
**Confidence:** 3

**Summary:**

The authors propose a generative model which assumes that the true data distribution is induced by a structural equation model (SEM). The proposed inference method uncovers both the underlying DAG structure of the SEM, as well as the generating functions in each of the DAG nodes, thus enabling the generation of synthetic data from the model. A number of numerical experiments show that the proposed model is competitive against the baselines.

**Strengths:**

+ An interesting approach combining DAG learning and generative modelling (e.g. Wasserstein loss)
+ Theoretical results providing the motivation for the proposed learning objective
+ Multiple numerical experiments showing the competitive performance of the proposed method

**Weaknesses:**

I think the presentation could be improved, I struggled to understand some of the details of the proposed method (see the questions below). Also some notation is confusing and there are a few typos, here are some examples:

- What is X (without subscript) in Eq. (1)?
- What is W in Eq. (2)?
- Typo in Definition 3.2. (G_A \neq G_A)
- What does X = A^T X + Z mean in Section 5.1.? Is it a recurrence relation?
- What exactly is "all observations" in Corollary 3.2.1.?

**Questions:**

- Eq. (1) defines the SEM through the expectations. Does it mean that you consider all SEMs with the same expectations (but potentially different joint distributions) as the same SEM?
- Could you please provide more details on the network architecture in Sec. 4? As I understood the first layer (parameterised by W^1) learns the DAG structure, while the subsequent L-1 layers learn the functions f_i in the SEM. Is it correct? What is the architecture of these layers? How to you estimate the Wasserstein distance?
- What modifications of the baseline methods did you do when you say "To make a fair comparison, we add noise to each competitor’s architecture"? (in Sec. 5)
- Why do you think the proposed method outperforms Zheng et al. (2020) on structural discovery while using their architecture?

---

### Official Review · Reviewer_XSCR · 2023-10-30

**Soundness:** 2 fair
**Presentation:** 3 good
**Contribution:** 1 poor
**Rating:** 3
**Confidence:** 5

**Summary:**

The paper studies the DAG learning problem. Different from the previous method, the paper utilizes both MSE and MMD loss to measure the distance between generated and truth data distribution. Extensive experiments on both synthetic and real-world datasets demonstrate the effectiveness of the proposed method.

**Strengths:**

1. The DAG learning problem is important.
2. The proposed method achieves better results than the baseline.

**Weaknesses:**

1. The problem setting is unclear. It is unclear the major difference between the traditional DAG learning task with the studied DAG-based generative regression task.
2. The novelty of the paper is very limited. The major difference between the proposed method with NOTEARS-MLP is the additional MMD loss. The contribution is marginal.
3. It lacks an analysis of the identifiability of used loss on the causal structure.

**Questions:**

1. Does the model can be used for regression prediction task?

---

### Official Review · Reviewer_F2tX · 2023-10-30

**Soundness:** 2 fair
**Presentation:** 2 fair
**Contribution:** 2 fair
**Rating:** 3
**Confidence:** 3

**Summary:**

The paper introduces a generative regression through the DAG model, that is, learning the Directed Acyclic Graph (DAG) through actual data, and then modeling the regression function from parent variables to child variables based on neural networks.

**Strengths:**

This paper is written clear, and the idea of construct generative regression based on DAG models is valuable for applications in machine learning fields.

**Weaknesses:**

In this article, I did not identify any notable advancements in the areas of causal structure learning, regression modeling, or the use of generative regression grounded in causal models.

**Questions:**

In Section 4 of the paper, the primary methodology is presented with limited detail and without a proper analysis of its validity. Additionally, I have reservations about the efficacy of solely using the MSE criterion to learn the underlying causal structure.

---

### Official Review · Reviewer_K8HK · 2023-10-31

**Soundness:** 1 poor
**Presentation:** 1 poor
**Contribution:** 1 poor
**Rating:** 3
**Confidence:** 4

**Summary:**

This paper proposes an approach to perform causal discovery.

### Review summary:
This work is below the bar for NeurIPS, since it lacks clarity, has poor motivation, is not well situated in the literature, and presents many conceptual inaccuracies. For these reasons, I cannot recommend acceptance. That being said, the empirical results do seem encouraging, which might mean that the authors are onto something. With a significant effort to understand the literature better and how their approach fits in it, this work could be of practical interest.

**Strengths:**

- Many experiments with apparently very strong performance.

**Weaknesses:**

### **Confusing motivation/framing**
The introduction makes an analogy between regression and DAG learning, but the rest of the paper is only about DAG learning. Also, I couldn’t understand the contribution from the introduction.

### **I can’t understand the proposed approach**
The writing is in general very confusing. At first, $\tilde{X}$ was described as the “fake sample” generated from the learned model. But (2) and (3) suggest these are sampled conditionally on actual data samples X. Right? I cannot make sense  of the proposed approach. The lack of clarity casts a shadow over the apparently very strong performance of the approach in the experimental section.

### **Novelty is unclear**
The paper does not contrast clearly their approach with the rest of the literature. From what I understand of the method, the novelty is very low. The empirical results are impressive, but there is no compelling story as to why there is such an improvement.

### **Poor sentence formulations**
- In abstract: “We learn DAG by reconstructing the model to…” Should add “the” between “learn” and “DAG”.
- In intro: “... to identify independent variables (exogenous) that are directly associated with the dependent variable.” What does “directly associated” mean?
- In intro: “DAG was first introduced as part of the formal theories for causal graphs in epidemiology a very long time ago (Greenland et al., 2007), …” The phrase “a very long time ago” feels weird in the context of an academic paper.
- In intro: “Causality learning is more about discovering…” It’s the first time I read the term “causality learning”. It should be “causal learning”, no?

### **Inaccuracies/Imprecisions, sometimes major**
- In intro: “The goal of DAG-learning is to holistically capture interplays between the variables in an entire dataset, which extends standard regression analysis that hypothesizes a fixed many-to-one DAG structure between the independent and dependent variables (see more details in Section 3).” This seems inaccurate, since most regression methods allow for dependencies between the covariates.
- In intro: “Under the causal identifiability assumption (Neal, 2020), a causality model is capable of replicating the distribution of real data if and only if the model is correct.” What does it mean for a model to be “correct” here?
- In Section 3: you refer to || P(X) - P(\tilde X)|| as a “distance function”. But which one is it? The total variation distance? The Wasserstein metric? Unclear.
- Definition 3.1: What you call “DAG-based generative regression” is a very standard problem in the literature. Does it really warrant a definition?
- Lemma 3.1 is not really a lemma. It is simply a  vague claim that $\mathbb{E}||X - \tilde{X}||_2$ is a good quantitative measure of how far two distributions are. However it’s not, since it’s not even a metric. For instance, if X and \tilde X are identically distributed (I assume independence) then in general the expectation won’t be zero. Also, the “proof” doesn’t make much sense.
-  Section 3: “In addition to the Markov and faithfulness assumptions, the identifiability assumption assumes an identifiable causal model, which suggests that there is only one unique DAG structure to generate P(X).” I don’t understand this sentence.
- Definition 3.2 is unclear. Same for Lemma 3.2.
- I’m confused by Equation (2) and (3). Why are \tilde{X} and X related? I thought X was the data and \tilde{X} the generated data?

### **Inaccurate related work**
- In related work: “Recently, DAG-learning with continuous optimization framework NOTEARS (Zheng et al., 2018) has drawn much attention evidenced by a series of NOTEARS-based DAG learning methods (Yu et al., 2019); (Lachapelle et al., 2020); (Zheng et al., 2020), which have fundamentally transformed DAG-learning from combinatorial search into a solvable continuous optimization problem through the use of acyclicity constraints.” The phrasing here makes it sound like these continuous-constrained approaches have transformed the field. This is an overstatement. In a sense these methods are indeed fundamentally different from classical combinatorial approaches, but we cannot say that they have overthrown the older approaches.
- “Most of the existing methods involve only reconstruction (i.e. mean squared) loss, which is limited to the generative process with additive noise models.” This is not true, many methods do not make this assumption, like the classical PC algorithm and its variants, which are based on conditional independence testing. Another example would be DCDI [1], which uses normalizing flow (more expressive than additive noise models).

[1] Philippe Brouillard, Sébastien Lachapelle, Alexandre Lacoste, Simon Lacoste-Julien, and Alexandre Drouin. Differentiable causal discovery from interventional data. In Advances in Neural Information Processing Systems 33, 2020.

**Questions:**

See above.

---

### Official Review · Reviewer_sCo4 · 2023-11-04

**Soundness:** 2 fair
**Presentation:** 2 fair
**Contribution:** 2 fair
**Rating:** 3
**Confidence:** 4

**Summary:**

The paper proposes DAG-based generative regression learning the data generation mechanism from real data by explicitly involving DAG in the generative process. Experiments are condcted showing that their algorithm can outperform state-of-the-art methods in replicating the real data distribution.

**Strengths:**

1. The paper introduces a new approach, DAG-based generative regression for regression analysis, showing the potential to advance regression analysis by capturing the causal relations among variables.
2. Experiments are conducted on various datasets and results are provided in the form of metrics and visualization compared to SOTA methods .

**Weaknesses:**

1. The motivation of the model design is not clear, i.e, there is no evident theoretical basis to ensure that the model architecture can effectively capture the causal relationships among data.
2. The paper is not organized well enough to provide clear analysis.

**Questions:**

1. Can you provide more details about the experiments, such as the choice of evaluation metrics and the results on larger datasets?
2. How to prove that the algorithms proposed can capture the DAG structure in the data?